# Comparison of healthcare costs of patients with COPD on maintenance inhaled therapies between 2011 and 2019 in Hungary using a nationwide database

**Brigitta Dombai**[1☉], **Viktória Nagy**[2☉], **István Ruzsics**[3], **László Németh**[4], **Tamás Balázs**[4], **Balázs Sánta**[5‡], **Zsófia Lázár**[6‡]*

**1** Outpatient Health Care Services of Kispest, Pulmonology Centre, Budapest, Hungary, **2** Department of Pulmonology, Géza Hetényi Hospital of Jász-Nagykun-Szolnok County, Szolnok, Hungary, **3** 1st Department of Internal Medicine, Division of Pulmonology University of Pécs, Medical School, Pécs, Hungary, **4** Healthware Consulting Ltd., Budapest, Hungary, **5** Chiesi Hungary Ltd., Budapest, Hungary, **6** Department of Pulmonology, Semmelweis University, Budapest, Hungary

☉ These authors contributed equally to this work.
‡ BS and ZL also contributed equally to this work.
* lazar.zsofia@semmelweis.hu

## Abstract

### Introduction and objectives

Medical costs of patients with chronic obstructive pulmonary disease (COPD) are high, however data from Eastern European countries are scarce. We aimed to study healthcare payments for patients with COPD on maintenance inhaled therapy in Hungary and analyse the trends and influencing factors between 2011 and 2019 in a retrospective financial database analysis.

### Patients

We collected data of patients from the Hungarian National Insurance Fund, who were > 40 years old, received maintenance inhaled therapy > 90 days within 12 months prescribed for J41-44 International Classification of Diseases-10 codes. All-cause and COPD-specific healthcare costs were compared between 2011 and 2019. We used a generalized mixed regression model to analyse the effects of calendar years, age, sex, Charlson comorbidity index, status of incidence, annual duration of inhaled therapy, the number of COPD-related hospitalization and geographical regions.

### Results

We analysed the data of 227 254 patients. In 2019, cumulative all-cause and COPD-specific spendings reached 401.15 million and 118.14 million USD, respectively. Annual total and COPD-related costs per patient in 2011 vs. 2019 were

**Data availability statement:** All relevant data are within the manuscript and its Supporting Information files. The National Health Insurance Fund of Hungary provided data in an aggragated format in line with local regulations, so no individual data analysis was possible.

**Funding:** The author(s) received no specific funding for this work.

**Competing interests:** No authors have competing interests.

2707 ± 3598 vs. 3332 ± 4463 USD and 927 ± 1162 vs. 981 ± 1534 USD, respectively (mean ± standard deviation). The increase in all-cause costs was above, while the rise in COPD-related costs was below the Hungarian inflation rate. The costs of medication and inpatient care comprised of the highest payment segments. The number of COPD-related hospitalizations had the most significant effect on the expenditures, while comorbidity burden and spendings on inhaled maintenance therapy were related to all-cause and COPD-specific costs, respectively. Increasing age was associated to higher spendings, but women had lower costs.

## Conclusions

The costs of inpatient care and medication are responsible for the largest segments of healthcare spendings for patients with COPD. Prevention of hospitalizations due to COPD and the close follow-up of comorbidities can help reduce medical costs.

## Introduction

Chronic obstructive pulmonary disease (COPD) is a heterogeneous lung condition characterized by chronic respiratory symptoms due to airway and/or alveolar abnormalities causing persistent, often progressive airflow obstruction [1]. COPD affects approximately 10 percent of the adult population worldwide, and it is one of the leading causes of death with 3.22 million COPD-related mortality in 2019 [1,2]. Despite the availability of diagnostic means and an increasing range of therapeutic options, optimal management of the disease is still challenging [3]. Given an aging population and an expanding array of risk factors, a further increase in prevalence and mortality can be predicted [4]. These changes will be driven mainly by the increasing prevalence and mortality in developing countries, however the developed world will also be affected [5].

COPD also poses a major economic burden on high income countries and their healthcare systems. Although reported annual costs per patient vary between countries due to different study designs and patient populations or the cost types analysed, several authors concluded that the healthcare costs of patients with COPD are higher compared to those without the disease [6–8]. More specifically, medication costs represented the largest part among the payment types in South Korea, and all-cause hospitalization and exacerbations had the highest influence on costs [9]. In line with this, hospitalizations were the most important determinant of all-cause medical costs in patients with different degrees of severity in the US [7,10], and in-patient costs together with spendings on medication were the highest cost categories both in the US [11], France [8], Denmark [6,8,11]. In addition, a population-based survey of more than 4 000 patients across 12 countries found that the hospitalization expenses were the highest segment in France, Germany, Spain, the USA, and South Korea, but spendings on home oxygen treatment were the largest individual direct costs in Brazil, Russia, France, and Italy [12]. However, data on healthcare costs of COPD in Eastern Europe are lacking.

Thus, in a nationwide database we aimed to analyse the treatment costs of patients with COPD on maintenance inhaled therapy in Hungary between 2011 and 2019. Besides exploring the distribution and temporal variation of spendings for healthcare services, we also aimed to unravel the factors associated with these costs.

## Materials and methods

### Study design and data source

A longitudinal, retrospective financial database analysis was conducted, utilizing the National Health Insurance Fund (NHIF) database, spanning in the period from 01 January 2011–31 December 2019. The NHIF plays a central role in the Hungarian healthcare system, as the only public healthcare insurer. It has individual budgets for the following expenditures: inpatient care, outpatient care, medication and medical aids [13]. The NHIF database collects healthcare-associated data, linking it to the International Classification of Diseases (ICD-10) codes, and connecting it to individual patients via their insurance number [14–16] (details are available in the Supporting information).

Hungarian citizens with social security coverage become eligible for social security-covered NHIF care (specific coverage per payment segments relevant to COPD are detailed in the Supporting Information). Access to these services primarily depends on establishing a legal connection through work status, as insurance fees are generally automatically deducted from monthly salaries, so all people employed in Hungary are automatically eligible for insurance. Furthermore, all citizens under 16 years of age, and all full-time students (including high school, university or other educational institutions) below or above 18 years of age, and pensioners are insured. Finally, unemployed people are also eligible to apply for free insurance without social security coverage and are obliged to pay a healthcare service contribution fee. Medical services not covered by social security are available but represent a negligible share.

In the database of the NHIF, patient information is linked to individual patients by their insurance number, which are unique identifiers for all people in Hungary, provided at birth. This identifier is only used to access the healthcare data and health insurance status of the individual and not applicable for personal identification. The NHIF has the legal right to handle patients' data (Act No. 80/1997 on mandatory health insurance coverage) and is also allowed to share it on a claim basis (based on Act 63/2012 on the reuse of public data). As stated in the contract terms, it is not allowed to gain access to data on individual patients or medical results that come from aggregating the data of less than 10 individuals from the database. Statistical analysis was performed between 01 March 2023 and 01 March 2024. During this interval patients' aggregated data were collected from the NHIF's database, according to the approved study protocol. No data including date of birth, insurance number, number of any identification documents, or any individual patient data could be accessed that could identify individual participants during or after data analysis.

### Ethics statement

As a non-interventional clinical study on retrospective financial database analysis, it was approved by the National Institution of Pharmacy and Nutrition of Hungary, based on the favourable opinion of the Scientific and Research Ethics Committee of the Medical Research Council of Hungary (authorization number: OGYÉI/5071–1/2023). As stated in this approval, patient consent was not required and not obtained.

### Patients

Eligible patients were over 40 years old, being the major age group even in early COPD [17], who received COPD maintenance inhaled therapy, i.e., long-acting muscarinic antagonists (LAMA) or long-acting beta2-agonists (LABA) alone or in combination, or together with inhaled corticosteroids (ICS), prescribed with J41-44 ICD-10 codes (drugs are listed in Table 1 in S1 Appendix). Patients were enrolled starting from that year when a prescription was first filled and was followed by at least two additional prescriptions in the next 12 months (covering more than 90 days on therapy in the first relative year).

They were considered incident patients in the calendar year when the first prescription was filled in. To ensure exclusion of earlier misdiagnosed bronchial asthma, patients were excluded if more than 20% of their last 10 prescriptions were registered with J45 ICD-10 code (Fig 1). Further data were collected for each calendar year for the enrolled patients if the patients were alive on 1 January in the same year. Patients were analysed only in those calendar years when there was at least one J44 ICD-10 code registration.

## Data collection

Data regarding age, sex, place of residence, comorbidities, the number of days covered by maintenance inhaled therapy, number of COPD-related hospitalisations and medical costs reimbursed by the NHIF were collected. Charlson Comorbidity Index (CCI) [18] without COPD was calculated from data on comorbidities based on the methodology by Quan et al. [19]. Patients were classified into mortality risk groups using the CCI: 0–1: low-risk, 2–3: moderate-risk, ≥ 4: high-risk [18].

The first year of occurrence with J44 ICD-10 code was considered as the incidence year, and each subsequent year was regarded as a prevalence year if there was at least one J44 ICD code registration and non-zero all-cause costs in that year. The annual number of enrolled patients with zero costs was < 20 between 2012 and 2019. Since the dataset was available starting in 2011, we were unable to determine a patient's incident or prevalent status for that year. Overcoming this bias, we analysed the data only from 2012, when we could establish the patient status based on data from the previous year.

During cost analysis both all-cause and COPD-related (coded with J44 ICD-10 code) costs were studied. Costs on in- and outpatient care, medication, medical aids and long-term oxygen therapy (LTOT) were collected as these are separately reimbursed by the NHIF (more details available in the Supporting information). All costs were available in Hungarian Forint (HUF). In the study period, HUF-USD exchange rate varied, therefore we decided to present the costs on a fixed

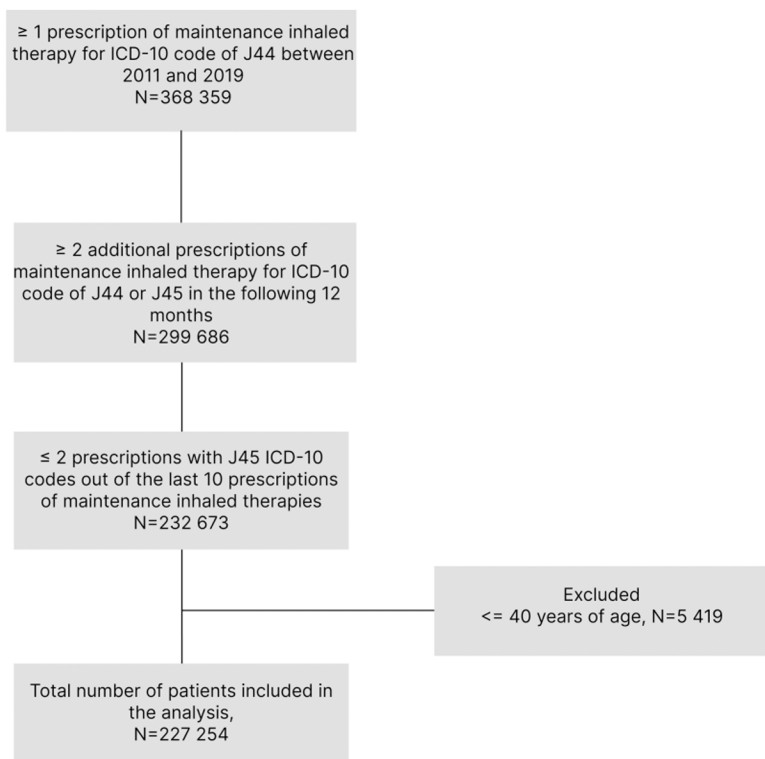

**Fig 1. Flowchart of patient enrolment.**

rate. All costs are presented in US dollars with the annual central rate of 2011 as set by the Hungarian National Bank (1 USD = 200.94 HUF) [20].

## Statistical analysis

Descriptive statistics were calculated for each calendar year using the variables mentioned above. For modelling the annual costs, a generalized linear mixed regression [21] was applied using Gamma distribution with logarithmic link function, which is considered to be the most suitable method for the analysis of right-skewed data as costs [22]. Covariates were age, sex, calendar year, status of incidence, place of residence, CCI, number of hospitalization due to COPD and number of days covered by maintenance inhaled therapy. The year 2011 was excluded from the modelling process to reduce bias of the incidence variable (as every patient was an incident patient in that year). Linear relationships between age and the number of hospitalizations or the days covered by maintenance inhaled therapy were considered too rigid for the model as a one-unit difference has generally less effect at larger values than at smaller values. As non-linear relationships were assumed for these covariates, quadratic terms were also used in the model, i.e., the linear and quadratic terms together determined the effect (multiplicative effect as shown in Figs 1-4 in S1 Appendix). The linear term had a stronger effect on estimates around zero, while the quadratic term was responsible for the behaviour of higher values. In case of a negative quadratic terms, the functions had peaks. The glmmTMB [23] and DHARMa [24] R packages were used for model fitting and diagnostics (Fig 5 in S1 Appendix).

Finally, a sensitivity analysis was conducted, where five models were compared with each other using Akaike information criterion (detailed in the Supporting information). The first model corresponded to the original model described above. In the second model the Charlson comorbidity index was replaced by individual comorbidities. A total of 27 comorbidities were analysed (Table 3 in S1 Appendix). Rare comorbidities (< 1%) and comorbidities with high multicollinearity (> 4), measured by variance inflation factor (Table 3 in S1 Appendix), were excluded from the analysis. In further models the following covariates were excluded from the original model: the number of maintenance inhaled therapy days, the number of hospitalizations or both. Due to computing capacity the sensitivity analyses were conducted on a representative sample of 20 000 patients. The results of the sensitivity analyses are shown in Table 4 in S1 Appendix. Model fit was best in the model, where individual comorbidities were also considered. However, the variables related to the calendar year showed similar behaviour here as in the original model using CCI. Excluding the number of maintenance inhaled therapy days or the number of hospitalizations resulted in worse fitting models, except for COPD-specific inpatient costs, where the fit of the original and the model without adjusting the therapy days is similar. Hence, we decided to report the original model with CCI, as it was also available in full sample size. The coefficients of the models with individual comorbidities obtained in the subsample are shown in Fig 6-9 in S1 Appendix.

## Results

### Patient characteristics and maintenance inhaled drug regimens

We analysed the data of 227 254 patients (Table 1). The annual number of patients increased in the study period, but this rise was lower than the corresponding number of incidence cases in that year. From 2017 the proportion of female patients exceeded that of males. There was a steady increase in age and comorbidity burden during the observation period.

Most patients used a single or a dual bronchodilator, the latter being available since 2015 in Hungary, with an increasing rate of patients on these therapies during the observation period. The proportion of patients using fixed dose ICS/LABA decreased over time, while as fixed triple combinations had become available in 2018, we saw a sharp increase in their use in 2019. The number of days on LABA or LAMA therapy exceeded that of ICS combination therapy in every year of the study period. The days covered with LAMA and LABA treatments were comparable, and from 2011 to 2019 the average number of days on treatment increased by 23.3% and 28.1%, respectively. Nonetheless, the days on ICS therapy decreased by 8.9%.

**Table 1. Demographic data and maintenance inhaled treatments of patients (total N = 227 254).**

| | 2011 | 2012 | 2013 | 2014 | 2015 | 2016 | 2017 | 2018 | 2019 |
|---|---|---|---|---|---|---|---|---|---|
| Total number of patients, N | 81 299 | 93 411 | 98 740 | 105 966 | 112 824 | 116 632 | 118 131 | 121 561 | 120 378 |
| Age, years | 66.6 ± 10.5 | 67.0 ± 10.4 | 67.4 ± 10.3 | 67.6 ± 10.2 | 67.9 ± 10.1 | 68.2 ± 10.0 | 68.4 ± 9.9 | 68.6 ± 9.8 | 69.0 ± 9.6 |
| Males, N | 42 388 (52.1) | 48 363 (51.8) | 50 787 (51.4) | 54 182 (51.1) | 57 010 (50.5) | 58 480 (50.1) | 58 788 (49.8) | 59 620 (49.1) | 58 312 (48.4) |
| Incident cases, N | NA | 22 185 | 19 159 | 20 284 | 20 382 | 17 942 | 16 342 | 17 112 | 12 548 |
| Charlson comorbidity index | 1.45 ± 1.62 | 1.55 ± 1.68 | 1.55 ± 1.69 | 1.59 ± 1.71 | 1.59 ± 1.71 | 1.60 ± 1.73 | 1.60 ± 1.72 | 1.60 ± 1.72 | 1.63 ± 1.73 |
| Number of hospitalizations due to COPD per patient, N | 0.19 ± 0.68 | 0.19 ± 0.68 | 0.19 ± 0.68 | 0.20 ± 0.7 | 0.20 ± 0.7 | 0.19 ± 0.69 | 0.19 ± 0.69 | 0.19 ± 0.7 | 0.18 ± 0.66 |
| The distribution of patients who filled in ≥ 1 prescription for maintenance inhaled drugs in that given year*, N: | | | | | | | | | |
| LAMA | 62 607 (77.0) | 67 853 (72.6) | 72 770 (73.7) | 81 960 (77.4) | 90 712 (80.4) | 95 681 (82.0) | 96 995 (82.1) | 100 345 (82.6) | 100 732 (83.7) |
| LABA | 64 802 (79.7) | 75 971 (81.3) | 80 183 (81.2) | 85 355 (80.6) | 91 746 (81.3) | 97 596 (83.7) | 99 152 (83.9) | 102 722 (84.5) | 102 895 (85.5) |
| LABA/LAMA FDC | 0 | 0 | 0 | 0 | 14 364 (12.7) | 33 081 (28.4) | 37 295 (31.6) | 40 927 (33.7) | 42 132 (35.0) |
| ICS/LABA FDC | 46 195 (56.8) | 48 086 (51.5) | 47 864 (48.5) | 48 694 (46.0) | 47 641 (42.2) | 47 568 (40.8) | 47 132 (39.9) | 47 228 (38.9) | 42 798 (35.6) |
| ICS/LABA/LAMA FDC | 0 | 0 | 0 | 0 | 0 | 0 | 0 | 4 031 (3.3) | 12 865 (10.7) |
| Number of days with maintenance therapy: | | | | | | | | | |
| LAMA | 163 ± 132 | 158 ± 141 | 158 ± 142 | 166 ± 139 | 173 ± 138 | 184 ± 139 | 188 ± 140 | 191 ± 139 | 201 ± 137 |
| LABA | 160 ± 126 | 171 ± 133 | 175 ± 135 | 171 ± 133 | 173 ± 134 | 186 ± 134 | 192 ± 136 | 196 ± 135 | 205 ± 134 |
| ICS | 123 ± 130 | 116 ± 135 | 112 ± 135 | 106 ± 131 | 101 ± 130 | 101 ± 132 | 102 ± 134 | 105 ± 135 | 112 ± 139 |

Data are shown as mean ± standard deviation (% of all patients). Abbreviations: FDC = fixed dose drug combinations, ICS = inhaled corticosteroids, LABA = long-acting beta2-agonists, LAMA = long-acting muscarinic antagonists, NA = not applicable.

*A patient is assigned to all maintenance inhaled drug categories if at least one prescription from that category is redeemed in that given year.

## All-cause and COPD-related healthcare costs

The annual average all-cause and COPD-related healthcare costs per patient in USD are shown in Tables 2 and 3. In 2019, the NHIF paid a total sum of 401.15 million USD for all healthcare services and 118.14 million USD for COPD-specific treatments of patients on maintenance inhaled therapy due to COPD. Between 2011 and 2019, there was a 23.1% increase in the average patient per year all-cause total healthcare costs, while COPD-related total spendings were unchanged until 2016, and only a 5.8%-rise in 2019 was noted in comparison with 2011. In line with this, the share of COPD-related costs per patient at average out of the total expenditures decreased from 34.2% in 2011 to 29.4% in 2019. Importantly, all items of costs showed a marked variation, proving high inter-individual differences.

Regarding all-cause costs, the highest annual expenditures per patient were spent on hospital care and medication in every year, however there was only a 5.8% increase in the costs of medication while the hospital care costs rose by 39.2% in 2019 compared to 2011. All payment groups showed an increase during the study period.

With respect to COPD-specific costs, medication and hospital care comprised of the highest payment segments in every year during the observation period, however, between 2011 and 2019 the average annual costs per person for medication decreased by 4.7%, while the spendings on in-patient care rose by 29.2%. Of interest, the annual average spending per patient on COPD-specific medical aids in 2019 was merely 3.9% of costs of all medical aids.

**Table 2. Annual all-cause healthcare costs per patient between 2011 and 2019 in USD.**

|  | 2011 | 2012 | 2013 | 2014 | 2015 | 2016 | 2017 | 2018 | 2019 |
|---|---|---|---|---|---|---|---|---|---|
| **Total** | **2707±3568** | **2595±3412** | **2607±2953** | **2664±3040** | **2704±3438** | **2864±3763** | **3096±4049** | **3246±4360** | **3332±4463** |
| Inpatient | 878±1985 | 900±2027 | 922±2087 | 937±2113 | 945±2165 | 1019±2353 | 1162±2670 | 1237±2816 | 1221±2776 |
| Outpatient | 248±273 | 258±277 | 263±280 | 277±292 | 282±304 | 309±348 | 356±386 | 383±416 | 391±427 |
| Medication | 1489±2566 | 1342±2379 | 1323±1595 | 1337±1661 | 1357±2198 | 1413±2440 | 1445±2463 | 1482±2741 | 1573±2909 |
| Medical aids | 91±382 | 95±379 | 100±400 | 111±430 | 120±452 | 123±463 | 133±490 | 145±517 | 146±519 |

Data are shown as mean±standard deviation. USD=United States Dollar.

**Table 3. Annual COPD-related healthcare costs per patient between 2011 and 2019 in USD.**

|  | 2011 | 2012 | 2013 | 2014 | 2015 | 2016 | 2017 | 2018 | 2019 |
|---|---|---|---|---|---|---|---|---|---|
| **Total** | **927±1162** | **879±1196** | **897±1276** | **882±1279** | **862±1253** | **889±1355** | **946±1532** | **969±1557** | **981±1534** |
| Inpatient | 236±1021 | 239±1069 | 251±1153 | 253±1159 | 252±1147 | 262±1252 | 309±1433 | 323±1457 | 305±1431 |
| Outpatient | 30±35 | 32±40 | 32±38 | 34±41 | 34±40 | 36±46 | 41±57 | 43±59 | 44±61 |
| Medication | 657±451 | 604±443 | 610±449 | 592±428 | 570±397 | 585±397 | 590±399 | 596±399 | 626±408 |
| Medical aids | 4±26 | 4±29 | 3±21 | 4±29 | 6±32 | 6±37 | 6±34 | 6±43 | 6±30 |
| LTOT* | 55±287 | 58±294 | 60±301 | 63±310 | 68±328 | 70±344 | 73±358 | 73±361 | 75±369 |

Data are shown as mean±standard deviation. USD=United States Dollar.

*The costs of long-term oxygen therapy (LTOT) are comprised of expenditures for medications and medical aids (more information is given in the Supporting information).

## Factors influencing all-cause total healthcare and inpatient care costs

The number of COPD-related hospitalizations, and the comorbidity burden had the highest direct effects on total all-cause healthcare costs (Fig 2A). Specifically, 1, 2, 3 and 4 hospitalizations were associated with 1.61-, 2.44-, 3.5-, and 4.79-times higher costs, compared to patients not requiring hospitalization (Fig 1 in S1 Appendix). As shown by the subsample analysis (Fig 6 in S1 Appendix), a wide range of individual comorbidities is linked with increasing costs. Among them, the effect of non-pulmonary cancer and metastatic cancer is the most prominent. Importantly, the increasing age of patients (Fig 1 in S1 Appendix), incident cases and certain residential regions in Hungary (Northern Great Plain and Southern Transdanubia) were associated with higher costs. Female patients had lower total and inpatient all-cause medical costs.

The increasing number of COPD-related hospitalizations had the largest direct effect on the costs of all-cause hospitalizations, but comorbidities and age were also associated with rising costs (at age 68 the multiplicative effect was 1.45; Fig 2B and Fig 2 in S1 Appendix). Among the comorbidities, cancer, metastatic cancer and acute myocardial infraction had the highest effect on inpatient costs (Figure 7 in S1 Appendix). The positive effect of age was stronger on all-cause healthcare costs than on COPD-related spendings (Figs 1-2 in S1 Appendix). Furthermore, incident cases and certain geographical regions (Southern and Central Transdanubia) were linked with higher inpatient care expenditures.

The observational year was associated with increased costs, especially from 2016, and this effect was more pronounced in 2019 on the expenditures on inpatient care than that on total costs.

## Factors influencing COPD-related total and inpatient care costs

As expected, the highest direct effect on total COPD-related costs were linked to the number of COPD-related hospitalizations with multiplicative effects being 2.6, 6.0, 12.0 and 22.0 at 1, 2, 3, and 4 hospitalizations (Fig 3 in S1 Appendix). Furthermore, costs were strongly related to the duration of maintenance inhaled therapy (Fig 3A), which was more prominent than in relation to all-cause healthcare costs (Fig 2A). We found a significant positive effect of age on COPD-related costs,

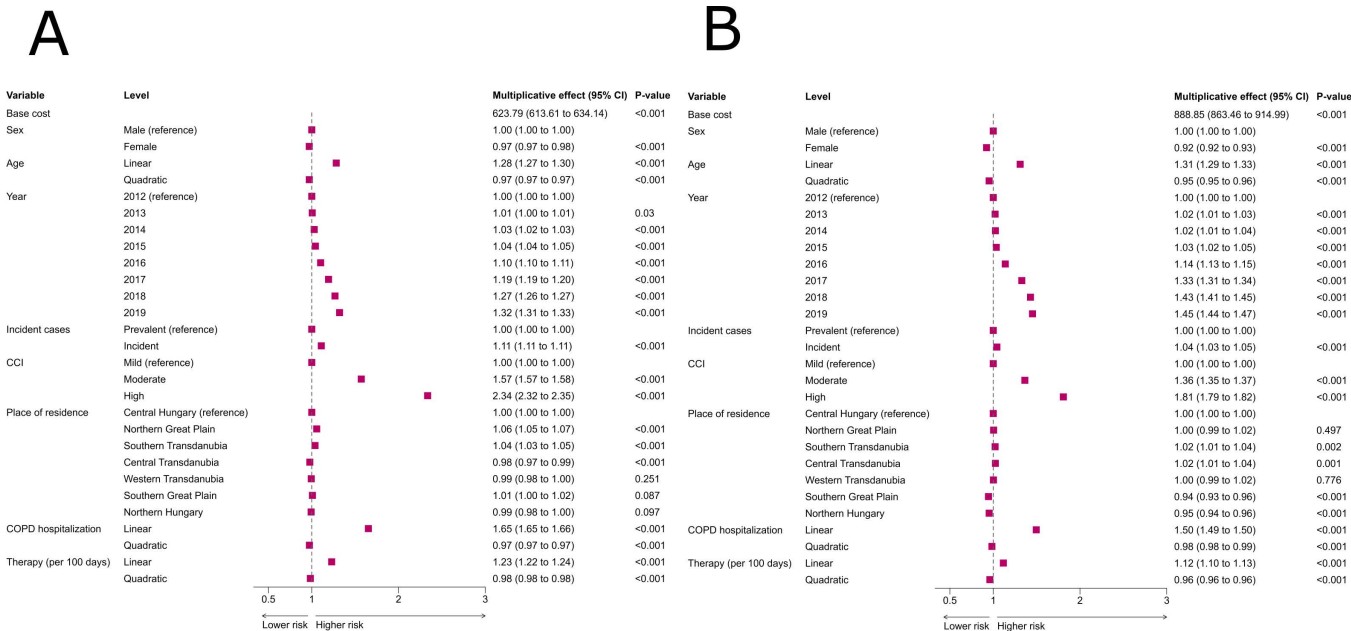

**Fig 2. The association of clinical variables to all-cause healthcare costs in COPD.** (A) Associations to total costs. (B) Associations to inpatient costs. A generalized linear mixed regression model was used. Abbreviations: CCI: Charlson comorbidity index, CI: confidence interval, COPD: chronic obstructive pulmonary disease, COPD hospitalization: the number of hospitalization for COPD in the calendar year, Therapy: inhaled maintance therapy.

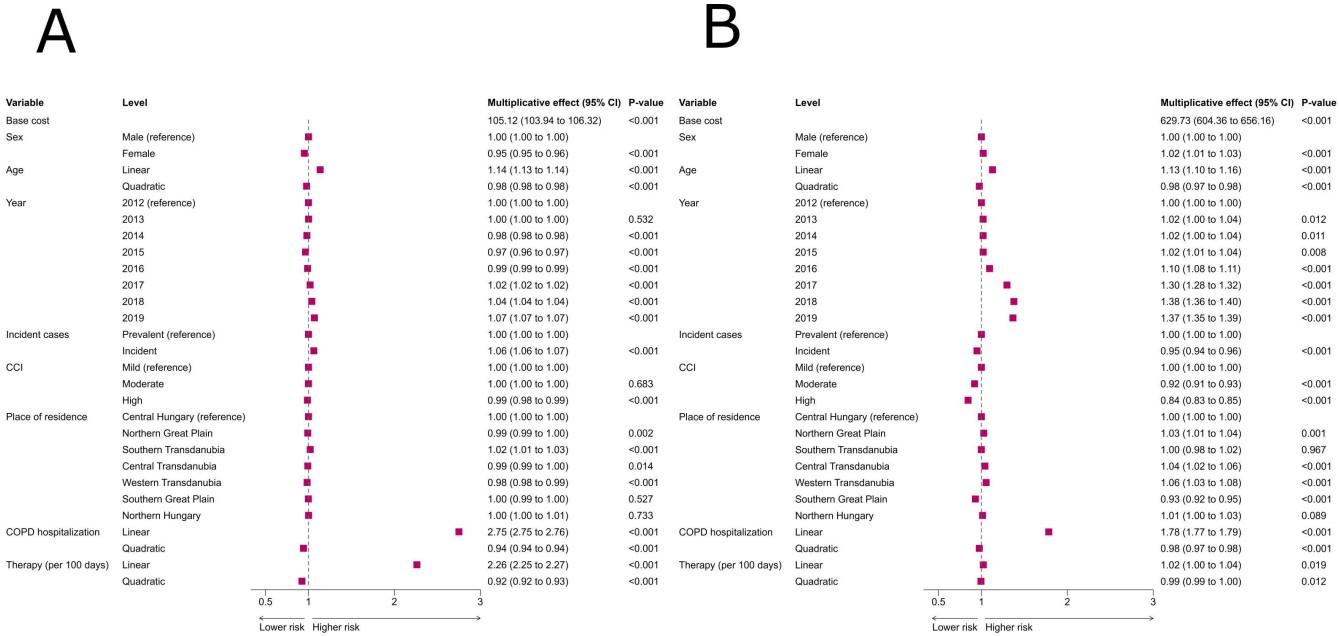

**Fig 3. The association of clinical variables to COPD-related healthcare costs in COPD.** (A) Associations to total costs. (B) Associations to inpatient costs. A generalized linear mixed regression model was used. Abbreviations: CCI: Charlson comorbidity index, CI: confidence interval, COPD: chronic obstructive pulmonary disease, COPD hospitalization: the number of hospitalization for COPD in the calendar year, Therapy: inhaled maintance therapy.

increasing till the age of 67 years (multiplicative effect: 1.20, Fig 3 in S1 Appendix). Female patients were associated with lower costs, and regional differences were also observed (Fig 3B).

The number of hospitalizations had the most prominent impact on COPD-related inpatient costs (Figs 3B and 4 in S1 Appendix). A higher number of comorbidities and incident cases were associated with lower expenditures. Interestingly, the presence of lung or metastatic cancers showed the most negative associations with costs among the comorbidities (Fig 9 in S1 Appendix). Patients till the age of 64 years showed a gradual increase in costs (multiplicative effect of 1.16 at 64 years, Fig 4 in S1 Appendix). Regional factors also contributed (higher costs in the Northern Great Plain, Central and Western Transdanubia, and lower costs in the Southern Great Plain).

We found a slight positive effect of the observational year on COPD-related total spendings starting from 2017, and a larger positive effect on COPD-related inpatient care costs (Fig 3), however, these were less than the effects on all-cause costs (Fig 2).

## Discussion

Healthcare costs of COPD are high worldwide; however, the disease still has high rates of morbidity and mortality, suggesting that healthcare spendings should be better tailored. Using a nationwide database of patients on maintenance inhaled therapies due to COPD in Hungary, we explored for the first time the healthcare spendings and the influencing clinical factors in the whole patient population. Between 2011 and 2019 there was a 23.1% increase in the total average all-cause healthcare costs per patient per year, while we found only a 5.8% rise in COPD-related spendings. The main determinant of expenditures per patient per year was the number of COPD-related hospitalizations, while age, comorbidities, sex and the duration of maintenance inhaled therapy were also linked with an effect on costs.

Between 2011 and 2019, the increase in the average annual all-cause total healthcare costs per patient merely exceeded the inflation rate in that period in Hungary (23.1% vs. 21.6%) [25], while COPD-related total spendings were unchanged until 2016, and only a 5.8%-rise in 2019 was noted in comparison with 2011. This might be partly explained by the increase in the reimbursement amount per Diagnosis Related Groups by the NHIF in 2017, however, it also demonstrates that health care spendings on COPD are lagging behind the treatment of comorbidities. Although COPD was the third leading cause of death in Hungary after cardio- and cerebrovascular and malignant diseases [26], if corrected with the inflation rate, COPD-specific healthcare spendings declined during this period, presumably contributing to the unfavourable statistics on COPD mortality. Importantly, the increase in COPD healthcare costs had been reported to be above the inflation rate between 2002 and 2012 in six countries of Europe and North America [12].

The largest proportions of costs were allocated to medication and inpatient care, which is in line with the results of previous studies. The average total medical costs per patient per year were similar to the figures in a study in 2009 from South Korea [9], however, higher average medical costs per patient per year were found between 2017 and 2018 in the USA [27], in 2012 in Germany [28], in 2004 in the United Kingdom [29], in 2005 in France [8] and in 2005 in Greece [30]. Our study is the first to report results from a nationwide data base analysis in an Eastern European country. A cohort study from Bulgaria focusing on patients with severe and very severe COPD showed higher direct average spendings due to COPD in 2015 (1372 Euro for medication and health care utilization) [31], which can be explained by the higher disease grade in their study compared to a general COPD population in our study.

In general, inpatient care costs more than outpatient healthcare services, and our results also corroborate previous findings on the importance of COPD-related hospitalizations on the costs [12]. In our study, both total all-cause and COPD-related spendings per patient per year were largely attributable to the number of hospital treatments due to COPD, supporting that the prevention of relapses should be one of the main aims of the treatment both from the perspectives of the disease course, symptoms, and the economic burden. Of note, COPD-related total costs comprised of only one third of the total all-cause annual spendings per patient.



COPD-related inpatient costs increased after 2016, which could not be related to the higher number of COPD hospitalizations as these were stable during the study period (Table 1), but it is probably linked to higher standardized amount per Diagnosis Related Groups reimbursed by the NHIF in the study period, or to other factors such as longer hospitalizations, more treatments, more advanced therapies or better access to hospital care as suggested by the influence of regions (Fig 3B). However, these effects should be further studied in more details.

Our data underscore the importance of adequate assessment and treatment of comorbidities. The comorbidity index showed a considerable rise during the observational period, and high comorbidity burden, especially acute myocardial infarction and extrapulmonary cancer, was associated with increasing all-cause total and inpatient care costs. This is line with the findings in other cohorts [28,32], showing that comorbidities have significant effects on direct total medical costs in COPD.

On one hand, women had lower total and inpatient all-cause medical costs. It contradicts the previous findings of a multicentre observational trial in Italy [33], which reported higher total costs for female patients with COPD. Furthermore, Wacker et al. also found that females have higher direct total all-cause medical costs in a cohort study [28]. On the other hand, women were linked with somewhat higher inpatient COPD-related costs. This supports the results of a previous study from our group showing that females are predisposed to a higher risk of repetitive exacerbations [15].

Age has been known to influence direct total medical costs of COPD [28]. In our study, the positive effect of age was more marked on all-cause healthcare costs than on the COPD-related spendings. This can be interrelated with the positive effect of the comorbidity burden on the non-COPD expenditures, but not on the disease-related healthcare costs.

Interestingly, while the incident cases were linked with higher total all-cause and COPD-related costs, they had lower COPD-related hospitalization spendings, which can be explained by the earlier disease stages at diagnosis when hospitalizations due to COPD are less frequent, but total spendings can be more pronounced due to the treatment of comorbidities. Of note, we categorized patients into incident cases starting in 2012 based on the year of inclusion and data collected from 2011 and later. However, patients who had been diagnosed before 2011 and had no healthcare spendings due to COPD in that year, could be misclassified as incident. Nonetheless, regarding the high symptom burden and the chronicity of the disease this bias is presumably limited.

We explored different segments of expenditures, previously not having been investigated. First, costs of medical aids related to COPD were outstandingly low in comparison with spendings on non-COPD specific items (Tables 2 and 3), however, we did not collect specifications on non-COPD medical aids. Importantly, the use of medical aids for COPD, such as devices facilitating expectorations or mitigating respiratory muscle weakness, should be encouraged on adequate indications, and potential factors relating to underusage such as lack of education of healthcare professionals or social background of patients, should be further studied. Furthermore, the average share of LTOT out of the COPD-related total costs was smaller than in previous studies [10,11,34,35]. However, the high variation in these data suggests that it can be a considerable sum for a subgroup of patients, which should need further attention in specific studies.

Our study has limitations. As we could not gather information on lung function; thus, the severity of COPD could not be assessed, which is known to influence healthcare utilization and medical costs [8–10,28]. We assessed medication use only by the prescriptions filled in, however, this does not ensure adherence to therapy [36]. Importantly, not all types of healthcare spendings in Hungary are affected by annual inflation rates of HUF or HUF-USD exchange rates, i.e., the full prices of the major inhaled maintenance medications in HUF did not increase between 2011 and 2019 (see Table 2 in S1 Appendix). Hence, to avoid interpretation bias we did not adjust the costs to inflation or currency exchange rates. Furthermore, access to healthcare in COPD are affected by several factors including lack of patient education, availability of social support, distance to health facilities, and attitude of healthcare providers [37], perceptions of ease of access, quality of relationship with the general practitioner or perceived disease severity [38]. In addition, the financial background of patients also limits the regular use of inhaled maintenance drugs, which consequently also influences exacerbation rates. Unfortunately, we were not able to explore the influence of these factors on the costs, as our database could not be enriched by these additional parameters from the data of the NHIF.



## Conclusions

Our study is the first to report the healthcare costs of patients treated with maintenance inhaled therapies for COPD using a nationwide database in an Eastern European country. The average annual total all-cause and COPD-related spendings per patient are lower than in other developed countries and the increase in payments for treating COPD was below the inflation rate between 2011 and 2019. The costs of inpatient care and medication comprise of the largest payment segments. Prevention of hospitalizations due to COPD and the screening, and proper treatment of comorbidities can help reduce medical costs.

## Supporting information

**S1 Appendix. Supporting information.**
(DOCX)

## Acknowledgments

The authors thank the Research & Data Analysis Department of Healthware Consulting for completing the database search and statistical analysis.

## Author contributions

**Conceptualization:** István Ruzsics, Tamás Balázs, Balázs Sánta, Zsófia Lázár.

**Data curation:** László Németh, Tamás Balázs.

**Formal analysis:** László Németh, Tamás Balázs.

**Investigation:** Zsófia Lázár.

**Methodology:** Zsófia Lázár.

**Supervision:** István Ruzsics, Balázs Sánta, Zsófia Lázár.

**Writing – original draft:** Brigitta Dombai, Viktória Nagy, Balázs Sánta, Zsófia Lázár.

**Writing – review & editing:** Brigitta Dombai, Viktória Nagy, Balázs Sánta, Zsófia Lázár.

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
