## [Decision Letter · Decision Letter 0]

15 Oct 2024

PONE-D-24-31266Comparison of healthcare costs of patients with COPD on maintenance inhaled therapies between 2011 and 2019 in Hungary using a nationwide databasePLOS ONE

Dear Dr. Lázár,

Thank you for submitting your manuscript to PLOS ONE. After careful consideration, we feel that it has merit but does not fully meet PLOS ONE’s publication criteria as it currently stands. Therefore, we invite you to submit a revised version of the manuscript that addresses the points raised during the review process.

We look forward to receiving your revised manuscript.

Kind regards,

Kuo-Cherh Huang

Academic Editor

PLOS ONE

Journal Requirements:

2. We note that your Data Availability Statement is currently as follows: “All relevant data are within the manuscript and in Supporting Information files.”

Please confirm at this time whether or not your submission contains all raw data required to replicate the results of your study. Authors must share the “minimal data set” for their submission. PLOS defines the minimal data set to consist of the data required to replicate all study findings reported in the article, as well as related metadata and methods (https://journals.plos.org/plosone/s/data-availability#loc-minimal-data-set-definition). For example, authors should submit the following data: - The values behind the means, standard deviations and other measures reported; - The values used to build graphs; - The points extracted from images for analysis. Authors do not need to submit their entire data set if only a portion of the data was used in the reported study. If your submission does not contain these data, please either upload them as Supporting Information files or deposit them to a stable, public repository and provide us with the relevant URLs, DOIs, or accession numbers. For a list of recommended repositories, please see https://journals.plos.org/plosone/s/recommended-repositories. If there are ethical or legal restrictions on sharing a de-identified data set, please explain them in detail (e.g., data contain potentially sensitive information, data are owned by a third-party organization, etc.) and who has imposed them (e.g., an ethics committee). Please also provide contact information for a data access committee, ethics committee, or other institutional body to which data requests may be sent. If data are owned by a third party, please indicate how others may request data access.

Additional Editor Comments:

Dear Dr. Lázár,

We appreciate your submission to PLoS ONE. I have received the review reports from two referees who have extensive expertise in the fields of health economics, pharmacology, and secondary data analysis. As you could see from their very detailed and expertly comments, both reviewers have serious reservations about your study, including fundamental methodological issues and statistical analysis concerns from both reviewers.

For example, “My main concern is that they didn't apply a discount rate to the costs. This means that they don't take into account cost inflation. This is a big bias and limits the results unless they are adjusted.” (Reviewer 1); “For the linear regression model: The days covered by inhaled therapy were included as covariates, however, it is not clear whether everything was included in general as therapy or by group of medication or by molecular used. – Please clarify, because this is a factor clearly related to costs.” (Reviewer 1); “Would recommend including patients for duration of follow-up once diagnosed, even if zero costs. This is due to the irreversible nature of COPD and potential to overestimate costs due to exclusion of milder cases.” (Reviewer 2); “Based on the results shown in table 2, the costs are not statistically significantly more? There is a very high SD. Not sure that the conclusions of higher costs can be made based on these findings.” (Reviewer 2) -- among other substantial concerns and suggestions.

In addition, please enhance the resolution and quality of both figures 2 and 3 and re-visit the adequacy of the keywords, as the reviewers meticulously pointed out.

With all that said, I would like to invite you to resubmit your work. Please respond to each comment of all the reviewers carefully and thoroughly. Please explain where you feel you cannot completely agree with reviewers’ suggestions. Thank you.

Kuo-Cherh Huang

Academic Editor

Reviewers' comments:

Reviewer's Responses to Questions

**Comments to the Author**

1. Is the manuscript technically sound, and do the data support the conclusions?

Reviewer #1: Yes

Reviewer #2: Partly

2. Has the statistical analysis been performed appropriately and rigorously? 

Reviewer #1: Yes

Reviewer #2: No

3. Have the authors made all data underlying the findings in their manuscript fully available?

Reviewer #1: Yes

Reviewer #2: Yes

4. Is the manuscript presented in an intelligible fashion and written in standard English?

Reviewer #1: No

Reviewer #2: Yes

5. Review Comments to the Author

Reviewer #1: PONE-D-24-3126

Comparison of healthcare costs of patients with COPD on maintenance inhaled therapies between 2011 and 2019 in Hungary using a nationwide database

Keywords: Financial Database (not a mesh term)

Introduction:

Line 59: COPD also poses a major economic burden on high income countries and their healthcare systems [1]: Please adjust the reference to be more specific. And be a little clearer regarding the economic burden on health systems, regarding the use of resources in COPD.

In general: Please be a little more specific regarding the reported economic burden: Is it more in medicines? What proportion? Or in clinical care? Or in oxygen?

Methods:

Please clarify who are the candidates to be insured by the public NHIF system, for readers who are not familiar with the Hungarian health system. – Is it the entire population of the country? The unemployed? Those without income?

Insurance number: Is it a unique national identification number? Is it used for other daily aspects? Or only health insurance?

Patients:

Line 102: Age over 40 or ≥ 40 years. Please clarify.

Line 103: Medications: It is worth clarifying which therapies were included, perhaps in a supplementary table. How were they linked to the DB? A unique code for the medications? Were short-acting bronchodilators also included? Were other therapies such as LABA/LAMA included? – ICS alone? Roflumilast? Etc?

Data collection:

Line 115: Is CCI calculated by default in the database record? Or was it calculated by the authors?

Line 117: Medical costs reimbursed by the NHIF? What costs are not covered by the system – Worth clarifying.

Line 124: All costs in 2011 dollars? Why not all with 2019 costs and a discount rate applied? Otherwise the costs would be estimated without taking inflation into account for almost a decade.

Statistical analysis:

For the linear regression model: The days covered by inhaled therapy were included as covariates, however, it is not clear whether everything was included in general as therapy or by group of medication or by molecular used. – Please clarify, because this is a factor clearly related to costs.

Results:

Line 149: They talk about lack of adherence. How can we be sure that it is a lack of adherence? Or could it be a bad diagnostic classification? Or failures in the delivery and access to therapy? Adverse reactions? Suspension of therapy by the treating physician?

It is interesting that an incremental cost is found in hospital and outpatient care. Although I do not see the number of hospitalizations in the results. It is worth including it. Is it due to a greater number of hospitalizations per patient? Or is it the same average but with an increase in costs. Please clarify, and for these reasons it is important to take into account price inflation for the analysis.

When looking at the figures, they are of low quality and very pixelated. They cannot be evaluated. Send in better resolution. To review correctly.

Discussion:

Include in the discussion why the increase in costs related to hospitalization? More or higher costs of these? And the potential reasons for this increase

Keeping in mind that, in theory, the main impact of current therapies is related to the reduction of the number of exacerbations. Or definitely patients are not continuously accessing therapies and it is a problem to be further explored in the health system.

Expand strengths and limitations.

Reviewer #2: In this study, authors examined the temporal trends in the costs of COPD in a Hungarian population from 2011 to 2019. While the findings are interesting, I do have the following suggestions, comments and concerns:

1. Not familiar with the phrase ‘which corresponds with a decline in real value for the latter’

2. Would recommend addition temporal findings in abstract (if space constraints, some of the methods can be concatenated)

3. Intro: International Classification of Diseases (IDC-10) should be ICD-10

4. Methods: Rationale for 40y/o+ can be added for readers not familiar with the disease

5. Methods: how were patients with co-existing asthma handled? Since it is a common comorbidity- it should be included and adjusted for

6. Methods: Would recommend including patients for duration of follow-up once diagnosed, even if zero costs. This is due to the irreversible nature of COPD and potential to overestimate costs due to exclusion of milder cases. Can be a supplemental analysis too.

7. Not sure I followed how every patient in 2011 is an incident patient. Would the prevalent users not be captured?

8. I do not think CCI can fully capture the comorbidity burden and recommend using individual comorbidities instead.

9. Why is number of days of therapy included in the model? Wont adjusting for this variable lead to an overestimation of costs?

10. Why is number of hospitalizations included in the model? This is an outcome variable (through hospitalization costs)- as demonstrated in findings.

11. Based on the results shown in table 2, the costs are not statistically significantly more? There is a very high SD. Not sure that the conclusions of higher costs can be made based on these findings.

12. Figure 2 and 3 are very blurred- are they based on the regression coefficients? Is statistical significance considered.

13. Discussion section is hard to follow- more detailed comparison with other studies (including other countries ) in terms of costs, comorbidities, and factors affecting costs is needed.

6. PLOS authors have the option to publish the peer review history of their article (what does this mean? ). If published, this will include your full peer review and any attached files.

**Do you want your identity to be public for this peer review?** For information about this choice, including consent withdrawal, please see our Privacy Policy .

Reviewer #1: No

Reviewer #2: No

---

## [Author Response · Author response to Decision Letter 0]

6 Dec 2024

Dear Prof. Kuo-Cherh Huang,

Thank you for the opportunity to review our manuscript. We assessed all points raised by the reviewers and responded to all comments in a point-by-point manner, please see below.

Reviewer #1: PONE-D-24-3126

Comparison of healthcare costs of patients with COPD on maintenance inhaled therapies between 2011 and 2019 in Hungary using a nationwide database

Comment: Keywords: Financial Database (not a mesh term)

Response: Thank you, we reviewed the term “Financial Database” with “Database Management Systems”, which is listed as a MeSH term.

Introduction:

Comment: Line 59: COPD also poses a major economic burden on high income countries and their healthcare systems [1]: Please adjust the reference to be more specific. And be a little clearer regarding the economic burden on health systems, regarding the use of resources in COPD. In general: Please be a little more specific regarding the reported economic burden: Is it more in medicines? What proportion? Or in clinical care? Or in oxygen?

Response: Thank you for the observation, we restructured this paragraph in the Introduction to more specifically report the findings of previous studies. We also added a new reference (Foo et al. Continuing to Confront COPD International Patient Survey: Economic Impact of COPD in 12 Countries), which compared data on direct costs of COPD, including those on home oxygen treatment, in 12 countries using data from surveys.

Methods:

Comment: Please clarify who are the candidates to be insured by the public NHIF system, for readers who are not familiar with the Hungarian health system. – Is it the entire population of the country? The unemployed? Those without income?

Response: NHIF covers all Hungarian citizens who pay health insurance, which is generally automatically deducted from monthly salaries, so all people who are employed in Hungary are automatically eligible for insurance. Also, all people under 16 years of age, and all full-time students (including high school, university or other educational institutions) below or above 18 years of age, all pensioners are insured. Finally, unemployed people are also eligible to apply for insurance for free. Based on this we added further details to the Methods (lines 88-97) and the Supporting Information.

Comment: Insurance number: Is it a unique national identification number? Is it used for other daily aspects? Or only health insurance?

Response: Thank you for the insightful comment. Insurance numbers or social security numbers are unique identifiers for all people in Hungary, provided at birth. This identifier is only used to access the healthcare data and health insurance status (eligible for treatment or not, based on insurance paid) of everyone, it is not applicable for unique identification of a person. We specified this aspect in the revised manuscript (lines 86-98).

Patients:

Comment: Line 102: Age over 40 or ≥ 40 years. Please clarify.

Response: We collected data from patients older than 40 years of age, this was also corrected on Figure 1 as exclusion ≤ 40 years of age.

Comment: Line 103: Medications: It is worth clarifying which therapies were included, perhaps in a supplementary table. How were they linked to the DB? A unique code for the medications? Were short-acting bronchodilators also included? Were other therapies such as LABA/LAMA included? – ICS alone? Roflumilast? Etc?

Response: The exact type of medication (drug and brand names) is recorded on the prescriptions which are linked to the patients’ insurance number. If a patient fills that prescription in, the medication is linked to the patient within the NHIF database. Only maintenance treatment of COPD was assessed in the present study, meaning that short-acting bronchodilators were not included. As discussed in the first paragraph of the Results and in Table 1, LABA and LAMA monotherapies, combinations and their combination with ICS were also included. ICS and roflumilast monotherapies are not recommended as maintenance options for COPD treatment, hence they were not studied. Also, roflumilast is not available in Hungary, therefore it was totally excluded from the study. We added a new table (S1 Table) to the Supporting information listing all drugs and their brand names which were available as inhaled maintenance therapy for J41-44 ICD-10 codes in Hungary between 2011 and 2019.

Data collection:

Comment: Line 115: Is CCI calculated by default in the database record? Or was it calculated by the authors?

Response: CCI was calculated by the authors, based on the methodology by Quan et al. (Quan H, Sundararajan V, Halfon P, Fong A, Burnand B, Luthi JC, Saunders LD, Beck CA, Feasby TE, Ghali WA. Coding algorithms for defining comorbidities in ICD-9-CM and ICD-10 administrative data. Med Care. 2005 Nov;43(11):1130-9. doi: 10.1097/01.mlr.0000182534.19832.83. PMID: 16224307). This was clarified in the revised manuscript, and the reference was added (lines 139-40).

Comment: Line 117: Medical costs reimbursed by the NHIF? What costs are not covered by the system – Worth clarifying.

Response: We fully agree with the reviewer. Healthcare costs are fully or partially covered by NHIF. All costs of out- and inpatient care, laboratory tests, imaging examinations are fully reimbursed, while certain percentage of costs of medication (in case of COPD inhaled maintenance medications - 90%, oxygen therapy – 100%) and medical aids (usually 50%) are covered. If medication is fully reimbursed by the NHIF, the patient still has to pay 300 HUF (0.77 USD as of Nov 2024) for filling in a prescription. This is detailed in the Supporting information of the revised manuscript.

Comment: Line 124: All costs in 2011 dollars? Why not all with 2019 costs and a discount rate applied? Otherwise the costs would be estimated without taking inflation into account for almost a decade.

Response: During the 2011-2019 period, inflation and fluctuations in the HUF-USD exchange rates could have influenced overall costs. In our analysis, we took multiple financial factors into account.

Firstly, we should note that the annual inflation rate in Hungary was relatively low during this period, with an average rate of 2.22%, and a cumulative rate of 23.1% between 2011 and 2019, as also discussed in the original and in the revised manuscript (lines 153-6). The highest inflation occurred in 2012 at 5.7%, which we used as the reference year in our model. Additionally, many basic therapeutical costs were price-controlled by the National Health Insurance Fund (NHIF), and these rates were not adjusted to the rate of inflation. Specifically, here we show that there was no increase in the total prices of the most frequently prescribed maintenance inhaled medications for COPD during the study period using publicly available information. Indeed, due to the appearance of generic drugs, the price of some drugs have declines (e.g. the LAMA tiotropium, which used to be marketed as the patented Spiriva Handihaler, but after the expiration of the patent the drug is now available in various formats of dry powder inhalers, leading to a decrease in the price of the original medication).

However, the standardized amount per Diagnosis Related Groups reimbursed by the NHIF increased in the study period.

Secondly, the HUF-USD exchange rate increased by over 50% between 2011 and 2019. If we had recalculated the USD value annually, it would have – misleadingly - shown a significant decrease in costs, largely due to the strengthening of the USD currency.

However, to avoid interpretation bias we did not adjust the costs to inflation or currency exchange rates. Considering these factors, we decided to present the original cost values without adjusting for inflation. To provide context for readers, we converted these values to USD, using a fixed exchange rate. However, we mention this as a limitation of the study in the revised manuscript (lines 353-7).

Statistical analysis:

Comment: For the linear regression model: The days covered by inhaled therapy were included as covariates, however, it is not clear whether everything was included in general as therapy or by group of medication or by molecular used. – Please clarify, because this is a factor clearly related to costs.

Response: Days covered by inhaled therapy was a covariate, that measures the number of days the prescribed drug provided inhaled therapy for. We constructed a therapy vector from each reimbursed medication, based on the days of treatment (DOT) covered by the drug package (please see S1 Table in the revised manuscript), and interpreted it as the number of covered days. This was used as one covariant (scaled by 100 days).

Results:

Comment Line 149: They talk about lack of adherence. How can we be sure that it is a lack of adherence? Or could it be a bad diagnostic classification? Or failures in the delivery and access to therapy? Adverse reactions? Suspension of therapy by the treating physician?

Response: We agree with the reviewer that the sentence of “The annual number of patients increased in the study period, but this rise was lower than the corresponding number of incidence cases in that year, showing a lack of adherence to inhaled maintenance therapies” raises concerns. We have reassessed this issue and acknowledge that lack of adherence is not an explanation for this phenomenon, as all patients who met the inclusion criteria of 3 months of inhaled maintenance therapy are consequently included in the study population irrespective of future adherence. However, those patients who are included but did not have medical costs in the given year were excluded from the cases. Importantly, the number of these cases was negligible as shown in the table below:

2012 2013 2014 2015 2016 2017 2018 2019

Number of patients with zero costs <10 <10 <10 16 16 15 14 11

Thus, the explanation for the increase in the study population being lower than the annual number of incident case is the mortality rate, we have rewritten the sentence in the Results. This was also mentioned in the Methods as “Further data were collected for each calendar year for the enrolled patients if the patients were alive on 1 January in the same year.” (lines 131-2).

Comment: It is interesting that an incremental cost is found in hospital and outpatient care. Although I do not see the number of hospitalizations in the results. It is worth including it. Is it due to a greater number of hospitalizations per patient? Or is it the same average but with an increase in costs. Please clarify, and for these reasons it is important to take into account price inflation for the analysis.

Response: Inpatient and outpatient expenditures are costs of all events per person in that certain year. We have added and extra row to the descriptive statistics table (Table 1) that shows the average hospitalizations of a person over the years to COPD. It seems that the 0.19 (sd 0.69) as the mean yearly number of hospitalizations did not change significantly over the years, therefore the cost increment should be explained by other factors (e.g. longer hospitalizations, increased costs, more treatments or more advanced therapies at the hospital). This was also added to the Discussion (lines 307-13).

Comment: When looking at the figures, they are of low quality and very pixelated. They cannot be evaluated. Send in better resolution. To review correctly.

Response: We improved the quality of the figures.

Discussion:

Comment: Include in the discussion why the increase in costs related to hospitalization? More or higher costs of these? And the potential reasons for this increase. Keeping in mind that, in theory, the main impact of current therapies is related to the reduction of the number of exacerbations. Or definitely patients are not continuously accessing therapies and it is a problem to be further explored in the health system. Expand strengths and limitations.

Response: Thank you for the comment. As shown above, the annual number of hospitalizations/patients because of COPD was relatively stable in the study period. Several factors can explain the link between the number of hospitalizations due to COPD and costs. Firstly, in-patient treatment in general costs more than outpatient care. The duration of hospital treatment and interventions during hospital stay also influence related healthcare costs. Specific to COPD, in the last 10 years, access to non-invasive ventilation as a treatment modality of COPD exacerbations with respiratory failure has greatly improved in Hungary with higher daily costs of therapy, possibly also contributing to higher overall costs (lines 307-13).

We fully agree with the reviewer that one of the main aims of the medical treatment of COPD is to prevent exacerbations. We also acknowledge that access to healthcare (both medical care and medication) is not homogenous in any countries which is also supported by our findings that certain regions in Hungary are associated with higher or lower costs of hospital care (both all-cause and COPD-related). Importantly, several factors have been described to influence COPD care, and thus maybe also costs, including lack of patient education, availability of social support, distance to health facilities, and attitude of healthcare providers (Nguyen TA et al. BMC Public Health 2021 DOI: 10.1186/s12889-021-11219-4), but perceptions of ease of access, quality of relationship with their general practitioner or perceived disease severity can also play a role (Shipman C et al. Prim Care Respir J 2009 DOI: 10.4104/pcrj.2009.00013). Unfortunately, we were not able to explore the influence of these factors on the costs, as our database could not be enriched by these additional parameters from the sources of the NHIF. We discuss these issues in the limitation of our study, and added the two references (lines 358-65).

Reviewer #2: In this study, authors examined the temporal trends in the costs of COPD in a Hungarian population from 2011 to 2019. While the findings are interesting, I do have the following suggestions, Comments and concerns:

Comment1. Not familiar with the phrase ‘which corresponds with a decline in real value for the latter’

Response: Thank you for the comment. We wanted to highlight that this increase in costs was less than the corresponding inflation rate, so we rewrote this sentence in the abstract.

Comment2. Would recommend addition temporal findings in abstract (if space constraints, some of the methods can be concatenated)

Response: Thank you for the valuable comment. The abstract was rewritten, and temporal findings were included.

Comment3. Intro: International Classification of Diseases (IDC-10) should be ICD-10

Response: We apologize for the typing mistake, we corrected it.

Comment 4. Methods: Rationale for 40y/o+ can be added for readers not familiar with the disease

Response: COPD usually develops over decades, mainly due to repeated exposure of harmful effects (e.g.: cigarette smoking, biomass burning, repeated pulmonary infections etc.). Hence, exclusion of younger patients can help reduce the number of misdiagnosis or at least, misclassified patients. Misclassification here means the incorrect use of ICD-10 codes due to administrative error. It has recently been shown in a population-based study focusing on patients < 50 years with early COPD, that almost 90% of patients in that age group were > 40 years old (Am J Respir Crit Care Med 2020 Mar 15;201(6):671–680). We added this information to the revised manuscript (line 121).

Comment 5. Methods: how were patients with co-existing asthma handled? Since it is a common comorbidity- it should be included and adjusted for

Response: Patients with concomitant asthma and COPD (COPD-A) were excluded, by limiting enrolment to patients who had most of their healthcare appearances with J44 code as compared to J45. A strict, absolute exclusion of the J45 code would limit our population too much, considering that it would exclude patients with misclassifications and those with difficult differential diagnosis especially at the beginning of the disease course. T

---

## [Decision Letter · Decision Letter 1]

16 Dec 2024

PONE-D-24-31266R1Comparison of healthcare costs of patients with COPD on maintenance inhaled therapies between 2011 and 2019 in Hungary using a nationwide databasePLOS ONE

Dear Dr. Lázár,

Thank you for submitting your manuscript to PLOS ONE. After careful consideration, we feel that it has merit but does not fully meet PLOS ONE’s publication criteria as it currently stands. Therefore, we invite you to submit a revised version of the manuscript that addresses the points raised during the review process.

We look forward to receiving your revised manuscript.

Kind regards,

Kuo-Cherh Huang

Academic Editor

PLOS ONE

Additional Editor Comments:

Dear Dr. Lázár,

Thank you for submitting your revised manuscript to PLoS ONE. Although the revised manuscript is much improved, the need for a couple of clarifications and revisions remains. Specifically, Reviewer 2 harbors doubts about your statistical analyses, and I believe that those issues are all rather essential and critical as the reviewer’s exhaustive comments pinpoint precisely. Please respond to each comment of Reviewer 2 carefully and thoroughly. Please explain where you feel you cannot completely agree with reviewer’s suggestions. Thank you.

Kuo-Cherh Huang

Academic Editor

Reviewers' comments:

Reviewer's Responses to Questions

**Comments to the Author**

1. If the authors have adequately addressed your comments raised in a previous round of review and you feel that this manuscript is now acceptable for publication, you may indicate that here to bypass the “Comments to the Author” section, enter your conflict of interest statement in the “Confidential to Editor” section, and submit your "Accept" recommendation.

Reviewer #1: All comments have been addressed

Reviewer #2: (No Response)

2. Is the manuscript technically sound, and do the data support the conclusions?

Reviewer #1: Yes

Reviewer #2: Partly

3. Has the statistical analysis been performed appropriately and rigorously? 

Reviewer #1: Yes

Reviewer #2: No

4. Have the authors made all data underlying the findings in their manuscript fully available?

Reviewer #1: Yes

Reviewer #2: (No Response)

5. Is the manuscript presented in an intelligible fashion and written in standard English?

Reviewer #1: Yes

Reviewer #2: Yes

6. Review Comments to the Author

Reviewer #1: The article was significantly improved, with adjustments by both reviewers. The quality of the figures and tables was improved. In general, the doubts and questions raised were answered.

Reviewer #2: The authors have addressed most comments, but the ones below still need work. For comment 8- VIF and tolerance can be tested for multicollinearity and knowing specific comorbidities and their association with costs is more valuable (of course higher CCI is related to higher costs- does not tell much). Specifically for comments 9 and 10, if the goal of the study is to estimate the costs, there is a need to reconsider the covariates as originally suggested or at the least, conduct sensitivity analysis- if authors still have access to the data. Alternatively, the focus can be just on factors affecting costs, and the trends piece/estimation can be removed as factors on the causal pathway for costs are included and adjusted for.

If the authors do not have access to the data anymore- I understand that they cannot carry out additional analysis - but they do need to clearly and explicitly discuss implications of these choices in the manuscript.

Comment8. I do not think CCI can fully capture the comorbidity burden and recommend using individual comorbidities instead.

Response: We fully agree with the reviewer that the use of CCI reduces the number of comorbidities that can be evaluated in the model. However, inclusion of all comorbidities in our analysis would not feasible, as their high number would reduce the intelligibility of the model. So, a selection of comorbidities is necessary. CCI is a well-known standardized predictor of mortality through all patient groups, and it could give valuable clinical associations to the outcomes of healthcare costs. However, to support that the set of comorbidities in in our study population is similar to other cohorts, here we list the 15 most common accompanying diseases in a cohort which was generated from the same database, and recently published by our group (Vincze K et al. Orv Het 2024, doi: 10.1556/650.2024.32981 - fulltext only available in Hungarian): hypertension, ischemic heart disease, bronchial asthma, anxiety, heart failure, cerebrovascular events (stroke, transitory ischemic attack), diabetes, osteoporosis, depression, arrhythmia (flutter, atrial fibrillation), anaemia, lung cancer, renal failure, aorta aneurysm (data not shown).

In our study the main aim was to find out the effect of calendar years costs for a patient. We used the GLM Gamma regression method, as it is considered as one of the best for population means (Malehi, A.S., Pourmotahari, F. & Angali, K.A. Statistical models for the analysis of skewed healthcare cost data: a simulation study. Health Econ Rev 5, 11 (2015).). In addition to the calendar years, we included common, well-known cost-related factors as covariates (age, sex, comorbidity, hospitalization, therapy length etc.). Some of these factors could also be potential outcomes, but our primary goal was to adjust with these covariates to make the effect of years more accurate. We decided to use CCI over separated comorbidities for statistical reasons. Using individual comorbidities, the risk of multicollinearity is very high, as comorbidities often occur together. This can cause instability and make the interpretation more difficult (i.e. cardiovascular disease and diabetes). Moreover, using too many covariates can result in overfitting in the model.

Comment: 9. Why is number of days of therapy included in the model? Wont adjusting for this variable lead to an overestimation of costs?

Response: Thank you for the comment. Number of days of therapy represents the adherence of patients to treatment. It is true that it is directly proportional to medication costs, but it can also have a strong effect on COPD-related outcomes, or even outcomes of comorbidities.

Days of therapy and the number of hospitalizations is known to increase medical cost, as every therapy has a price. We included these variables as explanatory factors in our models to account for these cost variations over the years. The model coefficients suggest that these factors are truly the most relevant ones, but the calendar years were in the centre of our focus, therefore adjusting is necessary. With this covariate, we aimed to include adherence into the model. We had two options: (1) using the days covered by therapies, or (2) using the number of prescriptions. We decided to use the first option as the same number of prescriptions can result in different therapy length as the days of treatment (DOT) per drug packages can be different (see S1 Table in the Supporting Information).

Comment10. Why is number of hospitalizations included in the model? This is an outcome variable (through hospitalization costs)- as demonstrated in findings.

Response: Thank you for raising this issue. Acute severe exacerbations were defined as hospitalisations or emergency ward visits with the ICD-10 code of COPD. This definition is not just used in financial database research but is also in line with current COPD guidelines. The number of severe exacerbations can increase costs not just through the cost of the COPD-related hospitalisations, but through the need for more medication and medical aid use for stable COPD and more resource spent on the control of comorbidities (as it is proven that the risk of worsening of multiple comorbidities is higher after an exacerbation). Taken all into account, we considered the inclusion of exacerbations necessary in the model.

7. PLOS authors have the option to publish the peer review history of their article (what does this mean? ). If published, this will include your full peer review and any attached files.

**Do you want your identity to be public for this peer review?** For information about this choice, including consent withdrawal, please see our Privacy Policy .

Reviewer #1: **Yes: ** Manuel E. Machado-Duque

Reviewer #2: No

---

## [Author Response · Author response to Decision Letter 1]

22 Feb 2025

Dear Prof. Kuo-Cherh Huang,

Thank you for the opportunity to review our manuscript. We assessed all points raised by the reviewer and responded to all comments in a point-by-point manner, please see below.

Reviewer #2: The authors have addressed most comments, but the ones below still need work.

Comment: For comment 8- VIF and tolerance can be tested for multicollinearity and knowing specific comorbidities and their association with costs is more valuable (of course higher CCI is related to higher costs- does not tell much). Specifically for comments 9 and 10, if the goal of the study is to estimate the costs, there is a need to reconsider the covariates as originally suggested or at the least, conduct sensitivity analysis- if authors still have access to the data. Alternatively, the focus can be just on factors affecting costs, and the trends piece/estimation can be removed as factors on the causal pathway for costs are included and adjusted for. If the authors do not have access to the data anymore- I understand that they cannot carry out additional analysis - but they do need to clearly and explicitly discuss implications of these choices in the manuscript.

Response: Thank you for the insightful comment.

We were able to access the original database, however, due to computational constraints, the additional sensitivity analysis could be carried out only on a representative subsample (20 000 patients, random sampling).

According to the suggestion of the reviewer, we collected the most relevant comorbidities also based on our previous publication (doi: 10.1556/650.2024.32981). In a single model, we aimed to determine their collinearities by VIF values. A total of 27 types of comorbidities were analysed (see below and also as S3 Table in the Supporting information). Rare comorbidities (< 1%) and comorbidities with high multicollinearity (> 4), measured by VIF were excluded from further analysis.

Individual comorbidities included in the model International Statistical Classification of Diseases and Related Health Problems 10th Revision (ICD-10) VIF

Acute myocardial infarction I21 1.068751616

Anaemia D50-D53, D55-D64 1.034723141

Anxiety F40-43 1.045229616

Asthma J45 1.024847372

Atrial fibrillation and flutter I48 1.071084957

Cancer C00-C76, C80-C97 1.614526599

Cerebrovascular diseases G45-G46, I60-I69 4.366433972

Cerebrovascular stroke I60-I69 4.339769592

Congestive heart failure I099, I110, I130, I132, I255, I420, I425-I429, I43, I50, P290 1.109509239

Connective tissue disorder M05-M06, M315, M32-M34, M351, M353, M36 1.007533029

Dementia F00-F04, F051, G30, G311 1.013943114

Depression F31-34 1.029095228

Diabetes E10-14, M14, N08 1.164134428

Diabetic complications E102-E105, E107, E112-E115, E117, E122-E125, E127, E132-E135, E137, E142-E145, E147 1.139024505

Hypertension I10-15, I67 1.098985732

Ischemic heart diseases I20-25 1.167677212

Liver disease B18, K700-K717, K73-K74, K76, Z944 1.005445392

Lung cancer C34 1.600050178

Metastatic cancer C77-C80 1.093350145

Osteoporosis M80-82 1.052310476

Paraplegia G041, G114, G801, G802, G81-G82, G830-G834, G839 1.006889935

Peptic ulcer K25-K28 1.008347355

Peripheral vascular disease I70-I71, I731, I738, I739, I771, I790, I792, K551, K558, K559, R02, Z958, Z959 1.025879615

Other pulmonary disease I278, I279, J40-J43, J46-47, J60-J67, J684, J701, J703 1.014078316

Renal disease I12, I13, N01, N03, N052-N057, N072-N074, N18, N19, N25, Z490-Z492, Z992 8.496715393

Renal failure N18-19 8.488893282

Sleep apnoea G4730 1.004138383

After finding the useful variables, we built and compared multiple models. Besides the version presented in the original manuscript, we replaced the CCI variable with individual comorbidities. In a third, fourth, and fifth model, we excluded the number of maintenance inhaled therapy days, the hospitalization number, and then both parameters from the original model, respectively. In order to find the statistically most relevant setup, we calculated the AIC for each model (also included as S4 Table in the Supporting information):

All-cause total cost All-cause inpatient cost COPD-specific total cost COPD-specific inpatient cost

Original model 1 370 963 561 377 1 119 811 151 013

Model with individual comorbidities 1 363 029 559 666 1 119 010 150 931

Model without adjustment to maintenance inhaled therapy days 1 373 959 561 658 1 172 853 151 010

Model without adjustment to hospitalization number 1 382 886 565 590 1 182 719 155 576

Model without adjustment to maintenance inhaled therapy days and hospitalization number 1 385 307 565 817 1 208 663 155 604

Model fit was best in the second model, where individual comorbidities were also considered. However, the variables related to the calendar year showed similar behaviour here as in the original model using CCI. Excluding the number of maintenance inhaled therapy days or the number of hospitalizations resulted in worse fitting models, except for COPD-specific inpatient costs, where the fit of the original and the model without adjusting to the therapy days is similar. We believe this supplementary analysis adequately addresses comments 9 and 10 raised by the reviewer, and do not support that the inclusion of maintenance inhaled therapy days and hospitalization number as co-variates. For comment 8, our additional analysis on a representative subsample shows that using individual comorbidities truly have a better fitted model.

Importantly, we add here the figures of the model with the individual comorbidities instead of the CCI. In line with the original results which showed that CCI is a major determinant of all-cause total and in-patient costs, we also found in the novel analysis that a wide range of comorbidities is linked with increasing costs. Among them, the effect of non-pulmonary cancer and metastatic cancer is the most prominent on the all-cause total and inpatient costs, while acute myocardial infarction shows also a marked positive associated with inpatients costs. However, regarding COPD-specific total costs the effect of comorbidities are much less, and there is a very modest positive association between costs and comorbidities (congestive heart failure, peptic ulcer, anxiety) which present symptoms potentially overlapping with those of COPD. Furthermore, no comorbidity increasing COPD-specific inpatients costs have been identified. Interestingly, concomitant diagnosis of cancer is linked with less COPD-related inpatient costs as COPD-related symptoms are possibly treated during hospital care primarily due to cancer treatment and not coded as hospitalization due to COPD.

However, in these models, the calendar years have similar effects as in our original model, therefore the message of our study is unaltered. As this analysis could not be performed in the whole data set, we kept the original figures in the main manuscript but expanded the section on “Statistical analysis” in the main text (lines 175-91), and in the Supporting information and included the results of this model in the Supporting information (S6-9 Figures) and mentioned them in the main text (lines 238-40, 253-5, 281-2, 338).

The association of clinical variables to all-cause total healthcare costs

The association of clinical variables to all-cause inpatient healthcare costs

The association of clinical variables to COPD-specific total healthcare costs

The association of clinical variables to COPD-specific inpatient healthcare costs

---

## [Decision Letter · Decision Letter 2]

27 Feb 2025

Comparison of healthcare costs of patients with COPD on maintenance inhaled therapies between 2011 and 2019 in Hungary using a nationwide database

PONE-D-24-31266R2

Dear Dr. Lázár,

We’re pleased to inform you that your manuscript has been judged scientifically suitable for publication and will be formally accepted for publication once it meets all outstanding technical requirements.

Kind regards,

Kuo-Cherh Huang

Academic Editor

PLOS ONE

Additional Editor Comments (optional):

Dear Dr. Lázár,

As nicely suggested by Reviewer 2, please add the additional analyses to the supplementary section. Thank you.

Kuo-Cherh Huang

Academic Editor

Reviewers' comments:

Reviewer's Responses to Questions

**Comments to the Author**

1. If the authors have adequately addressed your comments raised in a previous round of review and you feel that this manuscript is now acceptable for publication, you may indicate that here to bypass the “Comments to the Author” section, enter your conflict of interest statement in the “Confidential to Editor” section, and submit your "Accept" recommendation.

Reviewer #2: All comments have been addressed

2. Is the manuscript technically sound, and do the data support the conclusions?

Reviewer #2: Yes

3. Has the statistical analysis been performed appropriately and rigorously? 

Reviewer #2: Yes

4. Have the authors made all data underlying the findings in their manuscript fully available?

Reviewer #2: Yes

5. Is the manuscript presented in an intelligible fashion and written in standard English?

Reviewer #2: Yes

6. Review Comments to the Author

Reviewer #2: (No Response)

7. PLOS authors have the option to publish the peer review history of their article (what does this mean? ). If published, this will include your full peer review and any attached files.

**Do you want your identity to be public for this peer review?** For information about this choice, including consent withdrawal, please see our Privacy Policy .

Reviewer #2: No

---

## [Editor Report · Acceptance letter]

PONE-D-24-31266R2

PLOS ONE

Dear Dr. Lázár,

I'm pleased to inform you that your manuscript has been deemed suitable for publication in PLOS ONE. Congratulations! Your manuscript is now being handed over to our production team.

Kind regards,

on behalf of

Dr. Kuo-Cherh Huang

Academic Editor

PLOS ONE